# Multi-Frequency Light Sources Based on CVD Diamond Matrices with a Mix of SiV^−^ and GeV^−^ Color Centers and Tungsten Complexes

**DOI:** 10.3390/ma15238510

**Published:** 2022-11-29

**Authors:** Kirill V. Bogdanov, Ilya E. Kaliya, Mikhail A. Baranov, Sergey A. Grudinkin, Nikolay A. Feoktistov, Valery G. Golubev, Valery Yu. Davydov, Alexander N. Smirnov, Alexander V. Baranov

**Affiliations:** 1Center of Information and Optical Technologies, ITMO University, Kronverksky Pr. 49, bldg. A, 197101 St. Petersburg, Russia; 2Ioffe Institute, Polytechnicheskaya 26, 194021 St. Petersburg, Russia

**Keywords:** CVD diamonds, luminescent tungsten complexes, electron-vibrational coupling, temperature dependence

## Abstract

Recently, nanodiamonds with negatively charged luminescent color centers based on atoms of the fourth group (SiV^−^, GeV^−^) have been proposed for use as biocompatible luminescent markers. Further improvement of the functionality of such systems by expanding the frequencies of the emission can be achieved by the additional formation of luminescent tungsten complexes in the diamond matrix. This paper reports the creation of diamond matrices by a hot filament chemical vapor deposition method, containing combinations of luminescing Si-V and Ge-V color centers and tungsten complexes. The possibility is demonstrated of creating a multicolor light source combining the luminescence of all embedded emitters. The emission properties of tungsten complexes and Si-V and Ge-V color centers in the diamond matrices were investigated, as well as differences in their luminescent properties and electron-phonon interaction at different temperatures.

## 1. Introduction

The creation of multifrequency light-emission sources holds promise for various requirements of photonics and optical quantum technologies, and has potential applications in biomedicine and magnetometry. Diamond matrices with embedded color centers occupy a prominent position in the creation of these emission sources, due to such properties as biocompatibility, chemical resistance, and exceptional solidity [1]. Scientific interest in the study of diamond matrices with embedded color centers is supported by a wide range of methods for the synthesis of diamond matrices, and the possibility of introducing various atoms and complexes to tune their optical properties. To date, the most widely used methods of diamond fabrication are chemical vapor deposition (CVD), high-temperature and high-pressure (HTHP) techniques, and shock-wave synthesis [1]. The CVD method is considered optimal for the synthesis of diamond particles for the controlled introduction of impurity atoms, due to the possibility of creating a high-quality diamond matrix and controlling the number of impurity atoms.

The most popular and well-studied optically active defects are nitrogen-based, because of the simplicity of their synthesis and their exceptional spin properties. Such centers are particularly interesting for applications as quantum computers, sensors, and magnetometers. Unfortunately, their wide phonon-assistant component of luminescence and weak zero-phonon line (ZPL) as well as their impact on other centers means they are not the best option for constructing multi-frequency light sources. Optically active color centers based on embedded group IV atoms are nowadays the most popular emitters for such purposes. The main advantages of color centers based on group IV elements are the high intensity of emissions concentrated mostly in the ZPL, with low Huang–Rhys factors due to weak electron–phonon coupling and narrow FWHM (lower than ~5 nm at room temperature). The most common centers here are negatively charged silicon vacancy (SiV^−^) and germanium vacancy (GeV^−^) which have similar electronic structures with D_3d_ symmetry [2,3]. A few works devoted to these centers have analyzed their electron–phonon interaction and described the fine optical structure of the luminescent response and its dependence on the quality of diamond matrices. Thus, in several studies [4,5,6,7,8,9,10,11,12,13] of centers based on Si [4,5,6,7] and Ge [8,9,10,11,12,13], the temperature dependences of the positions and widths of the ZPLs have been described. The multifrequency emission sources with narrow zero-phonon luminescence lines at 602 nm and 738 for GeV^−^ and SiV^−^, respectively, have already been reported [11,14,15]. The process of improving the functionality of devices based on diamond matrices with embedded color centers is associated with extending the frequency range of the optical signal by the introduction of additional impurities. Therefore, the search for new color centers is of great importance, not only to extend the spectral range but also to find color centers with improved optical properties. To date, a huge number of possible variations of luminescent defects in the structure of diamond matrixes, including optically active variations, have been discovered [16,17]. The number of emission frequencies can be expanded by adding suitable impurities forming optically active centers into a diamond matrix that already contains, e.g., emitters based on SiV^−^ and GeV^−^ color centers. A promising idea appears to be embedding tungsten (W) atoms into a nanodiamond crystal lattice during CVD growth, to create luminescent tungsten-based defects. Indeed, there have been several reports of CVD fabrication of diamond crystal matrices with luminescing tungsten complexes that demonstrate intense emission bands in the near IR spectral range of 1.68–1.78 eV (705–750 nm) and are relatively transparent for biological tissues. Moreover, the simplicity of the implementation process of adding W complexes into the diamond matrix during CVD growth is particularly attractive [17,18,19,20]. It is well-established that the SiV^−^ and GeV^−^ centers are interstitial point defects of the D_3d_ point group symmetry, where the Si or Ge atoms are positioned between two adjacent vacancies along the <111> direction in the diamond lattice [3,7]. However, the position of the W complex in the diamond lattice has not yet been described. Following the Ludwig and Woodbury model of electronic structure [21] and based on obtained experimental data, researchers [18] have suggested that tungsten occupies an interstitial site. However, analysis of peculiarities of the local vibronic modes (LVM) spectra [19] allows us to assume that the W atom is a massive substitutional defect.

In this paper, we propose a variation on the expansion of emissions frequencies by additional doping of the diamond matrix with tungsten during the process of hot filament chemical vapor deposition (HFCVD) synthesis when tungsten atoms are captured by diamond matrices. Luminescent tungsten-based complexes have been formed producing very intensive zero phonon lines at ~714 nm (1.74 eV) and several intensive vibrational replicas with 24 meV energy distance. Therefore, we prepared and tested a multi-frequency light source based on HFCVD diamond matrices with emissions from the SiV^−^ and GeV^−^ color centers as well as tungsten complexes. It was shown that the incorporation of tungsten-based complexes into diamond matrices did not lead to significant changes in the structure and, accordingly, to changes in the optical response of color centers based on group IV atoms. The spectral properties of ZPL and vibration replicas of tungsten complexes were studied at a wide range of temperatures from room temperature to 7 K. The optical properties of tungsten complexes and color centers based on group IV atoms (GeV^−^) were compared. An evident distinction was revealed between temperature dependences of the positions and widths of emission lines for tungsten complexes and those for the color centers based on group IV atoms, mainly due to different electron–phonon interactions.

## 2. Materials and Methods

### 2.1. The Fabrication of Nanodiamonds with Luminescent GeV^−^, SiV^−^, and W Color Centers

The nanodiamonds were grown by the HFCVD technique from a methane–hydrogen mixture. Prior to HFCVD growth, detonation nanodiamonds with a characteristic size of ~4 nm, which served as nucleation centers, were deposited onto a germanium wafer by aerosol spraying [22]. The concentration of the detonation nanodiamonds on the substrate surface was about 107 cm^−1^. The parameters of the technological process included working pressure in the reactor of 50 Torr, hydrogen flow rate of 500 sccm, methane concentration of 2%, growth duration of 3 h, and the substrate holder temperature was 700 °C. The substrate holder was a molybdenum disk 2.5 mm in thickness and 20 mm in diameter. The temperature of the substrate was measured by a Pt–Pt/Rh thermocouple placed inside a hole in the substrate holder. The filament consisted of a six-turn coil constructed manually with 0.8 mm diameter tungsten wire. The filament was installed nonparallel to the substrate surface, and the distance between the filament and the substrate varied from 6 to 10 mm. The filament temperature was 2200–2300 °C, measured with an optical pyrometer. The tungsten impurities were incorporated into the nanodiamonds during the HFCVD process following tungsten evaporation from the filament [23]. Simultaneously, the filament heated the Si and Ge substrates. Final diamond size distributions were from 500 to 1000 nm.

The formation of SiV^−^, GeV^−^, and W complex in the course of HFCVD growth of nanodiamonds was carried out by introducing dopant atoms in the gas phase. Crystalline germanium substrate was used as a solid-state source of Ge atoms [15]. Residual silicon contamination on the substrate holder, due to the use of silicon wafers during HFCVD growth, was a source of Si atoms. The etching of Ge and Si atom sources with atomic hydrogen leads to the formation of the volatile GeH_x_ and SiH_x_ radicals, which move to the nanodiamond surface by means of the diffusion process. The tungsten impurities were incorporated into the nanodiamonds owing to tungsten evaporation from the filament [23]. Then, Ge, Si, and W atoms incorporated into the diamond lattice promoted the formation of SiV^−^, GeV^−^, and W complex in the diamond matrix. 

### 2.2. Scanning Electron Microscopy, Photoluminescence, and Cryogenic Measurements

The scanning electron microscope (SEM) images of diamond nanocrystals with formed luminescent GeV^−^ and SiV^−^ color centers and luminescent tungsten complexes were obtained with the Zeiss scanning electron microscope “Merlin” at an accelerating voltage of 10 kV and a probe current of 150 pA. Conventional approaches were followed to improve the image quality and topological contrast, namely, fixation of the samples with carbon tape to create a conductive bridge between the silicon substrate and the sample holder, and simultaneous registration of the signals with InLens and Everhart-Thornley SE2 detectors.

The photoluminescence (PL) spectra of the diamond nanocrystals in the temperature range of 7–273 K were measured using a LabRAM HREvo UV-VIS-NIR open spectrometer (Horiba, Lille, France) coupled with an RC102-CFM closed cycle helium cryosystem (Cryo Inc., Manchester, NH, USA). The excitation of the luminescence spectra was carried out using an Nd:YAG laser (Oxxius, Lannion, France) with continuous radiation λ = 532 nm (2.33 eV) focused using a Leica PL FLUOTAR 50× objective (NA = 0.55) onto a spot of diameter ~2 μm on the sample surface. The spectral resolution of the setup was ~2.5 cm^−1^ (~0.13 nm).

The spectrometer allowed simultaneous detection of the both the luminescence and Raman spectra of the samples used for control, by analysis of the diamond Raman line of ~1332 cm^−1^, and the crystal quality of the diamond nanoparticles with embedded impurity atoms forming the luminescent color centers. All measurements were taken at least five times to confirm the reproducibility of the data obtained.

## 3. Results and Discussion

Figure 1a presents the characteristic luminescence spectra of typical HFCVD diamond nanocrystal (nanodiamond) excited by 532.1 nm radiation at different temperatures, i.e., 7 K, 77 K, and 273 K. The ZPL peaks of GeV^−^ and SiV^−^ centers and W complex are seen in the vicinities of 602 nm [4,5,6,7], 738 nm [8,9,10,11,12,13,14,15,16], and 714 nm [17,18,19,20], respectively. The inset shows an SEM image of a typical HFCVD nanodiamond with a diameter of about 800 nm. The presence in the spectra of a narrow band at 572.6 nm corresponding to the Raman line of diamond at a frequency of 1332 cm^−1^ with a linewidth of ~5 cm^−1^ indicates the high quality of the diamond crystal lattice. Thus, the introduction of tungsten atoms sufficient to obtain an optical response did not significantly affect the structure of the diamond nanocrystal and did not lead to the appearance of a significant number of defects in the crystal lattice. 

Figure 1b,c shows the enlarged parts of the PL spectrum at a temperature of 7 K in the regions of GeV^−^ and W-complex emissions shown in Figure 1a. Lowering the temperatures to cryogenic levels led to the redistribution of the intensity between the luminescent lines, as well as to the appearance of a fine structure of ZPLs at the SiV^−^ and GeV^−^ centers. The fine structure of GeV^−^ is presented in Figure 1b, while the fine structure of SiV^−^ is barely visible in Figure 1c. It is well-known that single GeV^−^ as well as SiV^−^ centers have four separated ZPL bands corresponding to different electronic transitions [3,7]. Unfortunately, the presence of the centers within a different localization of matrices with different local strains [11,12] masks this structure and does not allow us to assign the peaks to the concrete electronic transitions. At temperatures lower than ~70 K a set of numerous narrow (~0.15 nm (0.5 meV) at 7 K) ZPLs of GeV^−^ centers become visible in the region near 602 nm, the positions of which differed from each other due to different local strains in the diamond nanocrystal [11,12]. A similar set of narrow ZPLs belonging to SiV^−^ centers was observed near 738 nm [24]. The fitting by Gaussians of the ZPL and vibration replicas of the local vibration mode (LVM) for W complexes are shown in panel (c). In the spectral region of 710–750 nm, the ZPL at 714 nm and a series of vibration replicas at 724 nm, 734 nm, etc., shifted in energy by 24 meV belonging to the LVM of W complexes [18,19] are observed. No set of narrow lines could be seen in the vicinity of the ZPL of the tungsten complex, even at lowest temperature of 7 K. The increase in temperature to higher than ~70 K led to homogeneous broadening of the ZPLs of SiV^−^ and GeV^−^ centers, resulting in the formation of heterogeneously broadened ZPLs for the ensembles of the centers, with Gaussian line shape and linewidth of about 5 meV. The ZPL and vibration replicas of W complexes underwent essentially less temperature broadening, but demonstrated more remarkable reduction in intensity. 

We noted important differences between the parameters of emission spectra of color centers based on W complexes and on group IV atoms (SiV^−^and GeV^−^):stronger temperature-induced reduction of the ZPL and LVM replicas;higher relative intensities of the vibration replicas with respect to that of ZPL, i.e., essentially higher values of the Huang–Rhys factor (S). Simple estimation of the Huang–Rhys factor defined by *I*_ZPL_/*I_tot_* = e^−*S*^ [25,26] gives a value rising from 0.9 to 1.5 in the temperature range 7–273 K, while S values of about 0.5–0.65 were reported for the SiV^−^ and GeV^−^ centers [2,25,26];significantly broader (~5–10 times) width of ZPL and LVM replicas at temperatures close to 0 K.

PL spectra of randomly selected single diamond nanocrystals were measured in the temperature range 7 K to 237 K. Figure 2 demonstrates a representative set of PL spectra of single HFCVD diamond nanocrystal in the range of W complex emissions, at several different temperatures. The spectra contain the ZPL (714 nm) of tungsten complexes and several LVM replicas (724 nm, 734 nm, 745 nm, etc.) with characteristic energy of 24 meV. The ZPL of the SiV^−^ center at 738 nm is visible in the low energy area of the tungsten’s second LVM replica.

To observe the processes governing the ZPL and LVM replicas in emissions of the W complexes, we obtained the temperature dependencies of the ZPL intensity and linewidth (FWHM) for the W complexes. In Figure 3 we show these dependencies for the temperature range of 7–273 K.

### 3.1. ZPL Intensity

Figure 3a presents the temperature dependence of the normalized ZPL intensity; the points represent the experimental data, and the solid line represents the fitting curve using a phenomenological calculation. Formula (1) was successfully applied for description of the reduction of the ZPL intensity in semiconductor heterostructures, SiV^−^ [27], NV [28] centers, and W complexes [19] in diamonds, due to phonon-assisted non-radiative processes:(1)$$\frac{I\left(T\right)}{I\left(0\right)}={\left(1+A\cdot \exp\left(-\frac{\Delta E}{k_{B}T}\right)\right)}^{-1},$$
where $I\left(0\right)$ is the ZPL intensity at 7 K, A is the negative constant, ΔE is the phonon thermal activation energy, and $k_{B}$ is the Boltzmann constant. It can be seen that experimentally observed decrease in the intensity of the ZPL of W complexes with increasing temperature was well fitted by Formula (1), supporting the conclusion that reduction in ZPL intensity is mainly due to the thermally activated nonradiative recombination mechanism. The calculated fitted value Δ*E* = (17 ± 1) meV was close to the LVM energy of 24 meV. This indicated that the LVM of the tungsten complex was mainly responsible for the temperature activation of the nonradiative relaxation of the excited state of the complex, competing with the process of radiative relaxation.

### 3.2. ZPL Linewidth

The large widths (~4 meV) of the ZPL and LVM bands even at 7 K, when homogeneous broadening induced by LVM dephasing the electron and vibronic transitions is relatively weak, may be caused by several factors. It is worth considering heterogeneous broadening due to the presence of several emission bands from different W isotopes with different transition energies, as well as homogeneous broadening due to the relatively high contribution of strong phonon-assisted (vibronic) transitions, and the increased rate of radiative relaxation of the W complex.

Isotopic effects were observed for GeV^−^ [29] and SiV^−^ [30]. A simple estimation of broadening of the ZPL and LVM bands due to isotopic shift was calculated according to the approach proposed by A. Dietrich et al. in framework of a simple harmonic oscillator model [30], allowing this variant to be excluded from consideration. Indeed, natural tungsten has four stable isotopes with a comparable content in the mixture: ^182^W (26.50%), ^183^W (14.31%), ^184^W (30.64%), and ^186^W (28.43%). The LVM band broadening can be roughly estimated by the energy shift between the LVM bands of the isotopes with heaviest and lightest masses: (2)$$\Delta E_{\mathrm{LVM}}\left(186,182\right)\sim E_{\mathrm{LVM}}\left(1-\sqrt{\frac{m_{186}}{m_{182}}}\right)=0.53\;\mathrm{meV},$$
where *E*_LVM_ of 24 meV is the energy of the experimentally measured LVM energy, *m*_186_ and *m*_182_ are the masses of the ^186^W and ^182^W isotopes. The same isotope-induced broadening value of 0.53 meV was estimated for the ZPL line. These values were almost an order of magnitude smaller than the measured FWHM of the ZPL and LVM peaks for W complexes, showing negligible contribution of the isotopic effect. We can therefore assume that at a temperature of 0 K the large ZPL width of the W complex is due to the relatively high contribution of phonon-assisted (vibronic) transitions increasing the rate of radiative relaxation of the W complex. This effect appears reasonable because of the high intensities of the LVM bands comparable with the ZPL intensity in the PL spectra of the W complexes with high Huang–Rhys value. 

Figure 3b shows the experimentally measured temperature dependence of the ZPL linewidth (FWHM) and its fitting by Formula (3):(3)$$Г\left(T\right)=\;{Г}_{R}+{Г}_{NR}+{Г}_{\mathrm{LVM}}\times \frac{1}{\exp\left(\frac{\hbar {Ω}_{V}}{k_{B}T}\right)-1},$$
where Г*_R_* is the homogeneous line width independent of temperature, due to the radiative decay of the excited state W in the complex via the ZPL and LVM channels, Г*_NR_* is responsible for homogeneous broadening due to the nonradiative relaxation of the excited state of the complex caused by the defects of host crystal lattice and surface, Г_LVM_ represents the intensity of electron-vibration coupling and is related to the Huang–Rhys factor (S), and *ħ*Ω*_V_* is the actual energy of the local vibrations. We propose in first approximation that in this formula the Г*_R_* and Г*_NR_* do not depend on temperature ${Г}_{R}+{Г}_{NR}=const.$ The fit used C, Г_LVM_, and *ħ*Ω*_V_* as the variable parameters, as shown in Figure 3b, and gave the following values: $C=\left(2.6\pm 0.2\right)\;\mathrm{nm}$, ${Г}_{\mathrm{LVM}}=\left(6.0\pm 0.3\right)\;\mathrm{nm}$, and $\hbar {Ω}_{V}=\left(22\pm 2\right)\;\mathrm{meV}$. The calculated fitted C value of 2.6 nm was close to the FWHM of ZPL at the lowest sample temperature of 7 K, while at 22 meV the value of *ħ*Ω*_V_* is almost equal to the LVM energy of the W complex, i.e., 24 meV. The latter indicates that the LVM of the tungsten complex is mainly responsible for the temperature-activated phonon dephasing the excited state of the complex, resulting in ZPL broadening. The fitted value of ${Г}_{\mathrm{LVM}}=6.0\;\mathrm{nm}$ was half that for the SiV^−^ center of 12.96 nm [27], which seems strange because electron-vibration coupling in the W complex was stronger than in the SiV^−^ center ($S_{W}\sim 1.25$ compared to $S_{SiV^{-}}\sim 0.6$). This fact and the temperature dependence of the S value in the W complex are yet to be studied and explained.

## 4. Conclusions

In this paper, we have reported the hot filament chemical vapor deposition synthesis of diamond nanocrystals containing a combination of luminescent SiV^−^ and GeV^−^ color centers as well as tungsten complexes. In addition to the well-known emissions of GeV^−^ and SiV^−^ color centers at 602 nm and 738 nm, a luminescent tungsten-based complex with an intensive zero phonon line at ~714 nm was accompanied by several intensive vibrational replicas with 24 meV energy distance. It was shown that the formation of tungsten-based complexes in diamond matrices did not lead to significant changes in the diamond crystal structure. The spectral properties of ZPL (position and linewidth) and the vibration replicas of tungsten complexes were studied at a wide range of temperatures from room temperature to 7 K and were compared with those of GeV^−^ color centers. It was shown that the W complex demonstrated high electron–phonon interaction with a Huang–Rhys factor of ~1.5, resulting in comparative intensity of ZPL and vibronic bands in the emission spectra of the complex. As a result, we demonstrated a multi-frequency light source based on HFCVD diamond matrices with emissions from both the SiV^−^ and GeV^−^ color centers as well as tungsten complexes.

## Figures and Tables

**Figure 1 materials-15-08510-f001:**
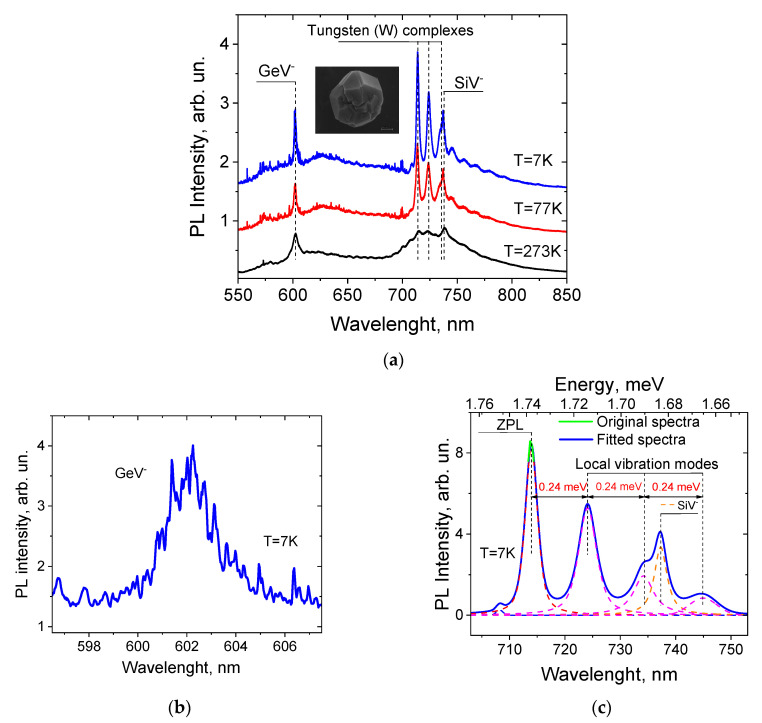
The PL spectra of the typical HFCVD nanodiamond under study. (**a**) Illustration of the temperature-induced evolution the PL spectra of GeV^−^, SiV^−^ centers and W complexes at temperatures of 7 K, 77 K, and 273 K; inset shows SEM image of a typical HFCVD nanodiamond with diameter of ~800 nm. (**b**,**c**) Enlarged parts of the PL spectrum at 7 K shown in (**a**) in the regions of GeV^−^ and W-complex emissions. The fittings of the ZPL and vibration replicas of W complexes by Gaussians are shown in panel (**c**).

**Figure 2 materials-15-08510-f002:**
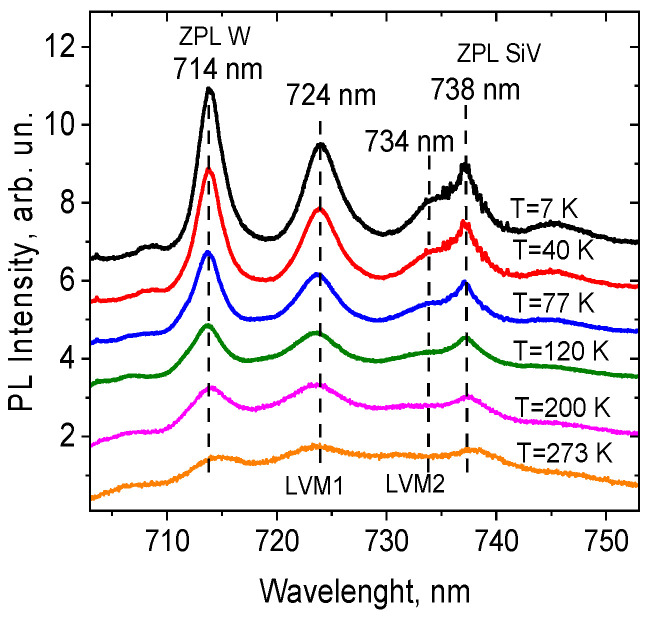
A representative set of PL spectra of single HFCVD diamond nanocrystal in the range of emission of W complex at different temperatures from 7 K to 273 K. Positions of the ZPL (714 nm) and several LVM replicas (724 nm, 734 nm, 745 nm, etc.) of tungsten complexes are shown. The ZPL of SiV^−^ centers at 738 nm can be observed in the low energy area of the tungsten’s second LVM replica.

**Figure 3 materials-15-08510-f003:**
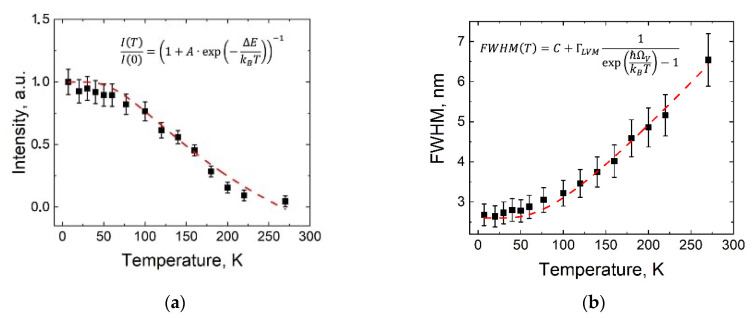
Temperature dependencies of the ZPL intensity and linewidth (FWHM) for the W complexes in HFCVD diamond nanoparticles. (**a**) ZPL intensity, (**b**) ZPL FWHM. Solid full squares represent the experimental data, the dashed lines are results of fitting with the least square method. Insets in the figures show the fitting formulas. The error bars are shown.

## Data Availability

Not applicable.

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
