# Peer review of "Multi-Frequency Light Sources Based on CVD Diamond Matrices with a Mix of SiV and GeV Color Centers and Tungsten Complexes"

_materials, 2022, doi:10.3390/ma15238510_

Round 1

Reviewer 1 Report

This manuscript by Bogdanov KV and colleagues provides a study, based on scanning electron microscopy and photoluminescence analysis at the ensemble level, of the relevant spectral features of GeV, SiV centers and tungsten complexes in diamond as a function of the temperature.

The main weakness of this work however is the lack of discussion of how the results compare with the existing data available in the literature.

The state of the art is finely and punctually discussed by the authors indeed; however the manuscript fails to highlight both the novelty and the differences in the results with respect to the state of the art would be highlighted.

In addition to the major concern stated above, I list here a few major issues to be addressed prior to any further considerations on the publishability of this work on materials:

1. we know that the four-line fine structure of GeV ZPL below 80K in Ref.[ Diamond & Related Materials 79, 145-149 (2017)]. Obviously, 7K is a relatively low temperature, so The author should write some text to illustrate this phenomenon. At the same time, “At temperatures lower than ~70 K a set of numerous narrow (~0.15 nm (0.5 meV) at 7 K) ZPLs of GeV centers become visible in the region near 602 nm, positions of which differ from each other due to different local strains in the diamond nanocrystal”, the numerous narrow should be marked in the fig 1(b) to help reader understand it better.

2. “The ZPL of SiV– center at 738 nm is seen at the low energy wing of tungsten second LVM replica” and “The ZPL of SiV– centers at 738 nm can be observed at the low energy wing of tungsten third LVM replica” sentences are incomprehensible. The series of vibration replicas is thought to be 724nm, 736nm, etc in line 169 in paper, however, 734nm and 745nm are also considered LVM replicas. I suspect that the reason is test temperature and it should be explained.

3. Important report [Appl. Phys. Lett. 177, 172104 (2020) ] indicate that temperature causes the NV center to quench. The phenomenon similar to Figure 2. Related introduction can be included for a more comprehensive background.

Author Response

Answer to Referee 1

First, we would like to thank the referee for the high evaluation of the work and valuable remarks and comments. We considered all the comments and believe that they made the text of the manuscript more understandable. Corrections made to the text are marked in red.

This manuscript by Bogdanov KV and colleagues provides a study, based on scanning electron microscopy and photoluminescence analysis at the ensemble level, of the relevant spectral features of GeV, SiV centers and tungsten complexes in diamond as a function of the temperature.

The main weakness of this work however is the lack of discussion of how the results compare with the existing data available in the literature.

We are pleased for the referee remarks and questions. Main problem for the W complexes is a lack of information about their structures and optic properties. In this work we correlated our data with several papers marked in our article [17-20]. We had shown well fitted descriptions for optical properties of W complexes by formulas 1-3 proposed by other authors. We also tried to show differences between well studied IV group centers and W complexes. Further studies and comparison will be completed by making some theoretical work in future papers.

The state of the art is finely and punctually discussed by the authors indeed; however the manuscript fails to highlight both the novelty and the differences in the results with respect to the state of the art would be highlighted.

In addition to the major concern stated above, I list here a few major issues to be addressed prior to any further considerations on the publishability of this work on materials:

  1. We know that the four-line fine structure of GeVZPL below 80K in Ref.[ Diamond & Related Materials 79, 145-149 (2017)]. Obviously, 7K is a relatively low temperature, so the author should write some text to illustrate this phenomenon. At the same time, “At temperatures lower than ~70 K a set of numerous narrow (~0.15 nm (0.5 meV) at 7 K) ZPLs of GeVcenters become visible in the region near 602 nm, positions of which differ from each other due to different local strains in the diamond nanocrystal”, the numerous narrow should be marked in the fig 1(b) to help reader understand it better.

 Thank you for this important clarification to the main text. The part related to fine structure of GeV and SiV ZPLs were added in a paper. We didn’t added peak's notations in the Figure 1(b) since it’s impossible to identificate the exact lines and most intense part are probably show just coincidence of same local strains of few centers.

  1. “The ZPL of SiV center at 738 nm is seen at the low energy wing of tungsten second LVM replica” and “The ZPL of SiV– centers at 738 nm can be observed at the low energy wing of tungsten third LVM replica” sentences are incomprehensible. The series of vibration replicas is thought to be 724nm, 736nm, etc in line 169 in paper, however, 734nm and 745nm are also considered LVM replicas. I suspect that the reason is test temperature and it should be explained.

Thank you for carefully reading our text. Unfortunately, there were the evident typos: the word “third” in the sentence in the Fig.2 caption has been replaced by “second”; the value of 736 nm related to one of the LVM replicas has been replaced by 734nm in line 169. In addition, we found loss of superscript (-1) in formula (1) both in the text and in Figure 3(a) that was been corrected.

  1. Important report [Appl. Phys. Lett. 177, 172104 (2020) ] indicate that temperature causes the NV center to quench. The phenomenon similar to Figure 2. Related introduction can be included for a more comprehensive background.

We agree with the Referee, the thermal quenching of the ZPL has, most likely, similar physical background for different color centers, including NV center reported in abovementioned Ref. [Appl. Phys. Lett. 177, 172104 (2020). For a better understanding of the general nature of the phenomenon, we have added a reference to this work in the sentence relating to the description of formula (1):

The sentence “Figure 3(a) presents the temperature dependence of the normalized ZPL intensity; the points represent the experimental data, and the solid line represent the fitting curve using a phenomenological expression (2) successfully applied for description of reduction of the ZPL intensity in semiconductor heterostructures and SiV centers [29] and W complexes [19] in diamonds due to phonon-assisted non-radiative processes:” has been replaced by

“Figure 3(a) presents the temperature dependence of the normalized ZPL intensity; the points represent the experimental data, and the solid line represent the fitting curve using a phenomenological expression (1) successfully applied for description of reduction of the ZPL intensity in semiconductor heterostructures, SiV [29], NV [30] centers, and W complexes [19] in diamonds due to phonon-assisted non-radiative processes:”

This reference as a #30 has been added to the reference list.

Reviewer 2 Report

The manuscript by K.V.Bogdanov et al. reports on production and spectral properties of negatively charged luminescent color centers based on atoms of the fourth group (SiV, GeV). This is an important and timely subject. The presented results are scientifically sound and well presented. Therefore, it is a pleasure to recommend the publication of the manuscript.

Before that happens, I would suggest that the authors should consider extension of their text by a short note on the NV- colour centers, which, while not belonging to the investigated fourth group, are probably the most studied color centers, so it would be instructive for the readers to compare their properties with those described in the manuscript.

Also, it would be interesting to know more about the charge stability of the centers studied. What about their ionization properties?

Finally, besides a short note in line 147 addressing the inset to Fig.1a, there is no information on the size distribution of the obtained nanodiamonds (in fact, for the crystalline size of 800 nm, it would be more appropriate to talk about micro- rather than nano-diamonds)

Author Response

Responses to Referee 2.

First, we would like to thank the referee for the high evaluation of the work and valuable remarks and comments. We considered all the comments and believe that they made the text of the manuscript more understandable. Corrections made to the text are marked in red.

Referee.

  1. The manuscript by K.V.Bogdanov et al. reports on production and spectral properties of negatively charged luminescent color centers based on atoms of the fourth group (SiV, GeV). This is an important and timely subject. The presented results are scientifically sound and well presented. Therefore, it is a pleasure to recommend the publication of the manuscript.

Before that happens, I would suggest that the authors should consider extension of their text by a short note on the NVcolour centers, which, while not belonging to the investigated fourth group, are probably the most studied color centers, so it would be instructive for the readers to compare their properties with those described in the manuscript.

Authors.

  1. Thank you for this important comment. We modified introduction and add important information about NVcolor centers.

Referee 2.

  1. Also, it would be interesting to know more about the charge stability of the centers studied. What about their ionization properties?

Authors.

2.Thank you for this question. It is well known that color centers based on embedded group IV atoms are negatively charged. The neutral charged centers based of group IV atoms are also exists. But we don’t see any PL features which can be evidence of presence such centers even at cryogenic temperature. Because of that and the fact that this is not the main topic of this article we don’t made experiments targeted to check their ionization properties.

Referee 2.

  1. Finally, besides a short note in line 147 addressing the inset to Fig.1a, there is no information on the size distribution of the obtained nanodiamonds (in fact, for the crystalline size of 800 nm, it would be more appropriate to talk about micro- rather than nano-diamonds)

Authors.

  1. Thank you for this important clarification to the main text. We added this information to the text of the paper in a materials section. As for the wording of size type, we are really using term nano-diamonds not because of any size dependant effect but because we are working with sub-micron size. Additionally we can reduce the size by ion-etching which probably will be shown in next papers.
